# Intra-Articular Leukocyte-Rich Platelet-Rich Plasma versus Intra-Articular Hyaluronic Acid in the Treatment of Knee Osteoarthritis: A Meta-Analysis of 14 Randomized Controlled Trials

**DOI:** 10.3390/ph15080974

**Published:** 2022-08-07

**Authors:** Yu-Ning Peng, Jean-Lon Chen, Chih-Chin Hsu, Carl P. C. Chen, Areerat Suputtitada

**Affiliations:** 1Department of Physical Medicine and Rehabilitation, Chang Gung Memorial Hospital at Linkou, College of Medicine, Chang Gung University, Guishan District, Taoyuan City 33343, Taiwan; 2Department of Physical Medicine & Rehabilitation, Chang Gung Memorial Hospital at Keelung, College of Medicine, Chang Gung University, Guishan District, Taoyuan City 33343, Taiwan; 3Department of Rehabilitation Medicine, Faculty of Medicine, Chulalongkorn University, King Chulalongkorn Memorial Hospital, Bangkok 10330, Thailand

**Keywords:** knee, osteoarthritis, leukocyte-rich platelet-rich plasma, leukocyte-poor platelet-rich plasma, hyaluronic acid

## Abstract

(1) Background: To evaluate the clinical effects of leukocyte-rich platelet-rich plasma (LR-PRP) and hyaluronic acid (HA) injections in treating patients suffering from knee osteoarthritis (OA); (2) Methods: Randomized controlled trials (RCTs) were searched from PubMed, Web of Science, and Cochrane Library. Keywords were: platelet-rich plasma, LR-PRP, leukocyte-rich, hyaluronic acid, and knee osteoarthritis. The included RCTs were published between the 1st of November 2011 and the 3rd of February 2021. Western Ontario and McMaster Universities Arthritis Index (WOMAC) scores, visual analog scale (VAS) scores, International Knee Documentation Committee (IKDC) scores, and adverse events were used as outcomes for evaluation; (3) Results: A total of 14 RCTs were enrolled. At 6 months, revealed that the LR-PRP group was better than the HA group in WOMAC total, pain, and physical function scores. At 12 months, the LR-PRP group was better than the HA group in WOMAC stiffness and physical function scores. There was no significant difference in adverse events; (4) Conclusion: LR-PRP injection showed no significant pain relief effect as compared with HA injection. However, LR-PRP demonstrated better overall outcomes as compared to HA in knee OA patients at the follow-up periods of 3, 6, and 12 months. LR-PRP injection may be recommended as a feasible option in treating patients with knee OA.

## 1. Introduction

Knee osteoarthritis (OA) is a common degenerative musculoskeletal disorder in the elderly population. Although total knee arthroplasty is proved as an effective solution for severe knee OA, the risks of operation complications have been mentioned. Thus, various non-surgical interventions have been described to treat symptomatic knee OA, including knee joint intra-articular (IA) injections, oral nonsteroidal anti-inflammatory drugs (NSAIDs), and physical therapy. The selection of IA injection treatments includes autologous platelet-rich plasma (PRP), corticosteroids, hyaluronic acid (HA), dextrose, and micronized dehydrated human amnion/chorion membrane [1]. IA application of low intensity radiation with specific wavelengths using Low Level Laser Therapy (LLLT) may be another non-surgical method that can be used in treating knee OA [2]. PRP is an autologous blood product containing growth factors, which include epidermal growth factor, fibroblast growth factor, platelet-derived growth factor, and vascular endothelial growth factor. These growth factors can improve angiogenesis, modulate inflammation, and recruit stem cells and fibroblasts to the area of injuries [3]. HA is a natural glycosaminoglycan in synovial fluid. HA can increase viscoelastic properties of the synovial fluid and increase overall joint lubrication in the injured region. It has been proven to facilitate functional improvements in knee, hip, and ankle OA [4]. In recent years, the efficacy of IA PRP and HA injections in knee OA patients has been compared in numerous studies. Dai et al. found that IA PRP injection is more effective than HA injection in pain relief and functional improvement at 12 months after the injection [5]. A meta-analysis conducted by Tang et al. indicated that IA PRP injection seems to be more effective than HA injection in long-term pain control of knee OA patients [6]. The efficacy of leukocyte-rich platelet-rich plasma (LR-PRP) and leukocyte-poor platelet-rich plasma (LP-PRP) has been controversial and without obvious consensus. Some preclinical studies have found that leukocytes may impair the overall effects of PRP [7]. Thus, further clinical trials are vital to compare the physiological effects of leukocytes in different PRP preparations. Belk et al. conducted a meta-analysis that showed that LP-PRP may be superior to LR-PRP in treating knee OA in terms of IKDC scores, but without significant differences in WOMAC and VAS scores [8]. Although multiple randomized controlled trials (RCTs) revealed that LR-PRP has better functional outcomes and pain relief than HA [9,10,11], no meta-analysis has solely discussed the efficacy of knee IA LR-PRP injection as compared with HA. This study aimed to perform a meta-analysis in investigating the effectiveness and safety of IA LR-PRP injection in the treatment of knee OA as compared with HA.

## 2. Results

### 2.1. Literature Search

Figure 1 demonstrates the literature selection process that was conducted in this study. During our initial literature search, 379 potentially eligible studies from PubMed, Web of Science, and Cochrane Library were found. A total of 167 duplicated studies were removed. The remaining 212 studies were meticulously screened according to the abstracts, and 194 of them were removed according to the following criteria: non RCTs, and those RCTs, which did not meet our inclusion criteria. Eighteen studies that had the potential for inclusion and their full texts were reviewed. Four studies were subsequently removed as they did not have control groups and data were not available. Finally, 14 RCTs were included in this systematic review and meta-analysis manuscript.

### 2.2. Main Characteristics of Included Studies

The main characteristics of included studies are listed in Table 1. In total, 14 RCTs were included in our study, with 1485 patients. A total of 815 patients underwent LR-PRP injections, and 670 patients underwent HA injections. These studies were published from 2011 to 2021. Five studies were conducted in Italy; two studies were conducted in Turkey and China, respectively. One study was conducted in Slovakia, Egypt, South Korea, Australia, and Iran, respectively. Further detailed characteristics and results of the involved patients are provided in Table 1. The dosage and the timing of LR-PRP and HA injections are shown in Table 2.

### 2.3. Quality of the Studies 

Figure 2 and Figure 3 presented the risk of bias. One study did not report methods of random sequence generation [11]. Eight studies reported allocation concealment [9,13,14,16,18,20,21,22]. Eight studies reported blinding of participants and personnel [9,10,12,13,14,16,20,21,22], and seven studies reported blinding of outcome assessors [9,10,16,18,20,21,22]. In total, five studies were double-blinded [9,10,16,21,22].

### 2.4. Outcomes of the Meta-Analysis

#### 2.4.1. WOMAC Total Scores (LR-PRP)

The comparison of LR-PRP and HA injections to the knee joints according to WOMAC total scores is shown in Figure 4. Due to the heterogeneity between trials being significant (*I*^2^ = 96%, *p* < 0.00001), the random-effect model was used. The combined analysis indicated that the LR-PRP injection was related to a decrease in WOMAC total scores as compared to the HA injection (MD = −3.96, 95% CI −6.17 to −1.75, *p* = 0.0004). A total of two RCTs compared WOMAC total scores at 1 month after injection (*I*^2^ = 78%, MD = −3.33, 95% CI: −8.84 to 2.18, *p* = 0.24) [17,19]; four studies reported WOMAC total scores at 3 months after treatment (*I*^2^ = 84%, MD = −2.7, 95% CI: −7.36 to 1.96, *p* = 0.26) [12,17,19,22]; four studies reported WOMAC total scores at 6 months after treatment (*I*^2^ = 95%, MD = −4.29, 95% CI: −7.66 to −0.91, *p* = 0.01) [12,17,19,22]; 3 studies reported WOMAC total scores at 12 months after treatment (*I*^2^ = 98%, MD = −6.45, 95% CI: −19.37 to 6.47, *p* = 0.33) [15,17,19]. We also performed the subgroup analysis and demonstrated that WOMAC total scores of the LR-PRP group were significantly different at 6 months after injection, compared to the HA group.

#### 2.4.2. WOMAC Pain Scores (LR-PRP)

Figure 5 demonstrated the comparison of LR-PRP and HA injections to the knee joints according to WOMAC pain scores. Due to the heterogeneity between trials being significant (*I*^2^ = 87%, *p* < 0.00001), the random-effect model was used. The combined analysis showed that the LR-PRP injection was related to a decrease in WOMAC pain scores as compared to the HA injection (MD = −0.58, 95% CI −0.97 to −0.2, *p* = 0.003). A total of two RCTs compared WOMAC pain scores at 1 month after injection (*I*^2^ = 58%, MD = 0.14, 95% CI: −0.68 to 0.96, *p* = 0.74) [17,19]; 4 studies reported WOMAC pain scores at 3 months after treatment (*I*^2^ = 90%, MD = −0.05, 95% CI: −0.93 to 0.83, *p* = 0.91) [17,19,22,23]; four studies reported WOMAC pain scores at 6 months after treatment (*I*^2^ = 42%, MD = −0.73, 95% CI: −1.05 to −0.41, *p* < 0.00001) [17,19,22,23]; 3 studies reported WOMAC pain scores at 12 months after treatment (*I*^2^ = 95%, MD = −2.17, 95% CI: −4.43 to 0.08, *p* = 0.06) [15,17,19]. We also performed the subgroup analysis and demonstrated that WOMAC pain scores of the LR-PRP group were significantly different at 6 months after injection, compared to the HA group.

#### 2.4.3. WOMAC Stiffness Scores (LR-PRP)

Figure 6 demonstrated the comparison of LR-PRP and HA injection to the knee joints according to WOMAC stiffness scores. Random-effect model was used as the heterogeneity between included RCTs was significant (*I*^2^ = 53%, *p* = 0.02). The combined analysis showed that the LR-PRP injection was related to a decrease in WOMAC stiffness scores as compared to the HA injection (MD = −0.33, 95% CI −0.52 to −0.14, *p* = 0.0007). A total of two RCTs reported WOMAC stiffness scores at 1 month after injection (*I*^2^ = 35%, MD = −0.11, 95% CI: −0.47 to 0.24, *p* = 0.53) [17,19]; three studies reported WOMAC stiffness scores at 3 months after treatment (*I*^2^ = 64%, MD = −0.15, 95% CI: −0.62 to 0.32, *p* = 0.53) [17,19,22]; 3 studies reported WOMAC stiffness scores at 6 months after treatment (*I*^2^ = 41%, MD = −0.29, 95% CI: −0.63 to 0.04, *p* = 0.09) [17,19,22]; three studies reported WOMAC stiffness scores at 12 months after treatment (*I*^2^ = 11%, MD = −0.65, 95% CI: −0.92 to −0.39, *p* < 0.00001) [15,17,19]. We also performed the subgroup analysis and demonstrated that WOMAC stiffness scores of the LR-PRP group were significantly different at 12 months after injection, compared to the HA group.

#### 2.4.4. WOMAC Physical Function Scores (LR-PRP)

Figure 7 demonstrated the comparison of LR-PRP and HA injections to the knee joints according to WOMAC physical function scores. Due to the heterogeneity between trials being significant (*I*^2^ = 83%, *p* < 0.00001), the random-effect model was used. The combined analysis showed that the LR-PRP injection was related to a decrease in WOMAC physical function scores as compared to the HA injection (MD = −2.58, 95% CI −3.69 to −1.48, *p* < 0.00001). A total of two RCTs reported WOMAC physical function scores at 1 month after injection (*I*^2^ = 59%, MD = −2.35, 95% CI: −5.27 to 0.58, *p* = 0.12) [17,19]; 3 studies reported WOMAC physical function scores at 3 months after treatment (*I*^2^ = 68%, MD = −1.03, 95% CI: −3.74 to 1.68, *p* = 0.46) [17,19,22]; three studies reported WOMAC physical function scores at 6 months after treatment (*I*^2^ = 0%, MD = −1.61, 95% CI: −2.34 to −0.89, *p* < 0.0001) [17,19,22]; 3 studies reported WOMAC physical function scores at 12 months after treatment (*I*^2^ = 93%, MD = −6.53, 95% CI: −12.13 to −0.94, *p*= 0.02) [15,17,19]. We also performed the subgroup analysis and demonstrated that WOMAC physical function scores of the LR-PRP group were significantly different at 6 and 12 months after injection, compared to the HA group.

#### 2.4.5. VAS Score (LR-PRP)

Figure 8 showed the comparison of LR-PRP and HA injection to the knee joints according to the VAS score. Due to the heterogeneity between trials being significant (*I*^2^ = 95%, *p* < 0.00001), the random-effect model was used. No significance of VAS score between LR-PRP injection compared to HA injection (MD = −0.02, 95% CI −0.35 to 0.30, *p* = 0.88) was found. A total of three RCTs reported VAS score at 1 month after injection (*I*^2^ = 0%, MD = 0.02, 95% CI: −0.12 to 0.15, *p* = 0.82) [16,17,19]; five studies reported VAS score at 3 months after treatment (*I*^2^ = 63%, MD = −0.15, 95% CI: −0.53 to 0.22, *p* = 0.43) [16,17,18,19,22]; 8 studies reported VAS score at 6 months after treatment (*I*^2^ = 95%, MD = 0.09, 95% CI: −0.57 to 0.75, *p* = 0.8); [10,11,13,14,17,18,19,22]; 4 studies reported VAS score at 12 months after treatment (I^2^ = 96%, MD = −0.36, 95% CI: −1.19 to 0.47, *p* = 0.4 [10,13,17,19]. We also performed the subgroup analysis and demonstrated that the VAS scores of the LR-PRP group showed no significance at 1, 3, 6, and 12 months after injection, compared to the HA group.

#### 2.4.6. IKDC Score (LR-PRP) 

Figure 9 showed the comparison of LR-PRP and HA injection to the knee joints according to the IKDC score. Due to the heterogeneity between trials being significant (*I*^2^ =70%, *p* = 0.0002), the random-effect model was used. The results showed that the HA injection was associated with a decrease in the IKDC score as compared to LR-PRP injection (MD = 3.58, 95% CI 0.61 to 6.54, *p* = 0.02). A total of three RCTs reported IKDC score at 2 months after injection (*I*^2^ = 0%, MD = −0.31, 95% CI: −2.94 to 2.32, *p* = 0.82) [9,10,13]; 5 studies reported IKDC score at 6 months after treatment (*I*^2^ = 67%, MD = 7.21, 95% CI: 3.01 to 11.41, *p* = 0.0008) [9,11,13,14,18]; 2 studies reported IKDC score at 12 months after treatment (*I*^2^ = 0%, MD = 2.43, 95% CI: −1.6 to 6.46, *p* = 0.24) [9,13]. We also performed the subgroup analysis and demonstrated that the IKDC score of the LR-PRP group was significantly different at 6 months after injection, compared to the HA group.

#### 2.4.7. Adverse Events (LR-PRP) 

Figure 10 compared the adverse effects between the LR-PRP and HA groups on knee OA in 8 RCTs [10,12,13,16,19,20,21,22]. The most common adverse events were worsening knee pain and swelling, vomiting, and fatigue. No major adverse events were reported in both groups. The heterogeneity test showed mild heterogeneity (*I*^2^ = 16%), thus the fixed-effects model was chosen. The results found no significance between LR-PRP and HA groups (RR: 0.86, 95% CI: 0.61 to 1.23, *p* = 0.41). The result indicated that LR-PRP and HA had similar safety profiles.

## 3. Discussion

Due to long life expectancy, the prevalence of OA in the elderly population has increased significantly. IA LR-PRP and HA injections have been widely used as the non-surgical treatment of knee OA [24]. This study collected 14 RCTs to investigate the efficacy of IA LR-PRP and HA injections in knee OA treatment. The results showed that LR-PRP significantly improved WOMAC total scores, WOMAC pain scores, WOMAC physical function scores, and IKDC scores at 6 months as compared to the HA group. At 12 months after IA injection, WOMAC stiffness scores and WOMAC physical function scores in the LR-PRP group were significantly better than in the HA group. No significant difference in adverse events was found between LR-PRP and HA groups.

There are various pharmacologic and interventional treatments to manage knee pain caused by knee OA and to prevent the need for knee joint arthroplasty. The non-operative treatment options included oral non-steroidal anti-inflammatory drugs (NSAIDs), activity modification, massage therapy, prolotherapy, IA corticosteroids injection, IA HA injection, IA PRP injection, and IA micronized dehydrated human amnion/chorion membrane injection [1,25]. Although previous evidence showed that IA PRP injection may improve the clinical conditions of pain and functional outcomes, several controversial issues remain, such as which is the best PRP preparation method, the interval between injections, the number of needed total injections, and the validity between LR-PRP and leukocyte-poor PRP (LP-PRP) [24].

Previously, many systematic reviews and meta-analyses have investigated the effects of PRP and HA in knee OA treatment [5,8,25,26]. Dong et al. compared the efficacy of PRP with other IA injections [26]. They found that IA PRP provided better outcomes (in pain relief and functional improvement) in knee OA patients, compared with other injection treatments, such as HA, saline, and prolotherapy. Dai et al. included 10 RCTs with 1069 patients and showed that IA PRP injection may have more benefits in WOMAC total, WOMAC pain, and WOMAC physical functions scores only at 1 year after injection as compared with HA and saline in treating knee OA patients [5]. However, our analyses showed that WOMAC total scores, WOMAC pain scores, and WOMAC physical function scores of the LR-PRP group were superior to the HA group at 3, 6, and 12 months after injection. Gong et al. demonstrated that PRP had significant advantages over HA in WOMAC total scores at 1, 6, and 12 months of follow-up. Our results also showed that WOMAC total scores were better in the LR-PRP group than in the HA group at 6 and 12 months after injection [25]. Tang et al. included 20 RCTs and found that PRP injection may relieve pain more efficiently than HA injection at 6 months and 12 months after treatment [6]. The PRP group also had better WOMAC total scores, WOMAC pain scores, and WOMAC physical function scores, as compared to the HA group at 6 and 12 months after treatment. Evaluated the IKDC scores at 3 and 6 months also showed that PRP injection was significantly more effective. Our results showed that IKDC scores, WOMAC total, WOMAC pain scores, and WOMAC physical function scores of the LR-PRP group were superior to the HA group at 6 and 12 months of follow-up. However, LR-PRP injection showed no significant difference in VAS scores at 6 and 12 months of follow-ups, as compared to HA injection. Belk et al. investigated 18 RCTs including 811 patients and found that IA PRP injection can be expected to improve clinical outcomes compared with HA in treating knee OA patients [8]. The analysis of the studies which compared the LR-PRP and the LP-PRP found no significant differences in the efficacy of WOMAC scores or VAS scores. However, the analysis indicated that LP-PRP may be superior to LR-PRP in IKDC scores.

Autologous platelet-rich plasma (PRP) can be divided into leukocyte-rich PRP (LR-PRP) and leukocyte-poor PRP (LP-PRP). By definition, LR-PRP is regarded as having a neutrophil concentration above baseline. LP-PRP is regarded as having a neutrophil concentration below baseline [27]. Present understanding is that LR-PRP may be associated with pro-inflammatory effects. Catabolic cytokines, such as interleukin-1β (IL-1β), tumor necrosis factor-α, and metalloproteinases, are increased in LR-PRP, which may have deleterious effects on chondrocytes [28]. LP-PRP on the other hand increases anti-inflammatory mediators of IL-4 and IL-10, which can further suppress the release of the inflammatory mediators of TNF-α, IL-6, IL-1β. Inflammation can be further blocked by neutralizing the nuclear factor-kB activity [29]. It seems that LP-PRP may be more suitable in treating knee OA due to its anti-inflammatory effect. However, there are also reports suggesting that the application LR-PRP may be a better choice in the treatment of knee OA. LR-PRP has a high percentage of mononuclear cell (MNC) recovery. Monocytes and macrophages are believed to play a critical role in the development and progression of knee OA. These innate immune cells guide vascular remodeling and recruit local stem cells, further stimulating the regenerative function of myeloid cells [30].

When it comes to the frequency of PRP treatment, most of the included trials received three consecutive LR-PRP injections [9,10,11,12,13,14,16,18]. In a further literature search, we have discovered four studies that received two LR-PRP injection treatments [15,17,19,21]. One study received four consecutive LR-PRP injections [20]. One study has mentioned the treatment effectiveness after receiving only one LR-PRP injection [22]. Due to insufficient trials, it is difficult to perform subgroup analysis to further arrive at the conclusion as to which frequency can offer the best treatment outcome.

The ideal composition, injection intervals, and injection times of PRP for knee OA injection treatment remain controversial. Previous studies did not find improved clinical outcomes when using LP-PRP to treat knee OA [10,17,23]. However, some studies have demonstrated that LP-PRP may be superior to LR-PRP in treating knee OA [9]. LR-PRP contains higher concentrations of growth factors and leukocytes and may be more painful when injected intraarticularly. LP-PRP may have a more anti-inflammatory effect as compared with LR-PRP as fewer proteases are released from white blood cells [9]. Further studies are needed to examine the influence of leukocyte concentrations in the PRP injectant on pain relief and functional outcomes in knee OA patients.

## 4. Materials and Methods

The meta-analysis of this study was performed based on the recommendations of the *Cochrane Handbook for Systematic Reviews of Intervention* [31] and the Preferred Reporting Items for Systematic Reviews and Meta-Analyses statement (PRISMA) [32]. IRB approval is unnecessary as this study is a systematic review of previously published RCTs that does not involve further processing of the patient data. The approved PROSPERO protocol number was CRD42022347244.

### 4.1. Systematic Search for Trials

RCTs were searched from PubMed, Web of Science, and Cochrane Library. MeSH terms and keywords were: leukocyte-rich, platelet-rich plasma, LR-PRP, hyaluronic acid, HA, knee osteoarthritis, and arthritis, which were used in various combinations. Reference lists and bibliographies of relevant systemic reviewers were also searched manually for every publication that may provide further information on PRP and HA studies. The included RCTs in our review were published between the 1st of November 2011 and the 3rd of February 2021. Initially, the two authors (Yu-Ning Peng and Jean-Lon Chen) independently extracted data by the same standard. The authors examined the full studies separately to determine whether they met our inclusion criteria. Discrepancies were resolved through discussion between authors.

### 4.2. Inclusion Criteria

We limited our search to RCTs related to humans. These RCTs were published in the language of English only. The RCTs were selected based on the following inclusion criteria: RCTs that compared IA LR-PRP injections with HA injections for the management of symptomatic knee OA; and adult patients with the diagnosis of knee OA.

### 4.3. Exclusion Criteria

Exclusion criteria were: adolescents or children (under 18 years of age); studies that were considered as cohort, case-controlled, cross-sectional, review article, and conference abstracts; RCTs without a control group; and cadaveric or animal studies.

### 4.4. Risk of Bias Assessment and Data Extraction

Two authors (Yu-Ning Peng and Jean-Lon Chen) independently evaluated each RCT according to the Cochrane risk of bias assessment scale [1]. Seven categories of bias were evaluated as the followings: blinding of participants and outcome assessment, random selection, reporting bias, allocation concealment, outcome data, and other study bias. Three levels were summarized in each category (high risk, low risk, and unclear risk). Two authors (Yu-Ning Peng and Jean-Lon Chen) independently extracted the following information from each included trial: first author’s name, publication year, country of origin, study type, number of patients, age, gender, and outcome measurements, and follow-up periods. We also collected injection time intervals and dosage of LR-PRP and HA injections. All the data were extracted from the RCTs from tables and texts.

### 4.5. Data Analysis

The systematic review and meta-analysis were conducted by Review Manager 5.3 (Nordic Cochrane Center, Cochrane Collaboration, Copenhagen, Sweden). We used mean differences (MD) and standard deviation (SD) to compare continuous variables. All data were reported with 95% confidence intervals (CI). We set the level of significance as *p* < 0.05. Higgins *I*^2^ statistic was used to evaluate the heterogeneity of individual studies. The fixed-effects model was utilized if no obvious heterogeneity existed (if *I*^2^ < 50%). Otherwise, a random-effects model was utilized (if *p* < 0.1 and *I*^2^ > 50%).

## 5. Limitations and Conclusions

There are some limitations to this study. First, all the included trials in this study were published in English, which may lead to the possibility of selection bias. Second, most of the analyses showed high heterogeneity. Although we tried to minimize the heterogeneity by performing subgroup analyses, high heterogeneity can still be observed in some results. This may be caused by the heterogeneity among patients, including gender, age, and the severity of knee OA. Other influential factors include the difference in PRP injection technique between studies, such as the frequency of PRP injections, injection volume, and injection intervals. Finally, some RCTs’ relatively small patient sample sizes limited the study power.

In conclusion, this is the first meta-analysis that demonstrated that IA LR-PRP injection showed better overall outcomes as compared to IA HA injection in patients with knee OA at 3-, 6-, and 12-month follow-up periods, in terms of WOMAC pain scores, WOMAC physical function scores, WOMAC total scores, and IKDC scores. LR-PRP injection offers a better overall treatment outcome as compared with HA in knee OA patients. As a result, IA LR-PRP injections offer better outcomes in treating knee OA patients, as compared with HA during both short-term and long-term follow-up periods.

## Figures and Tables

**Figure 1 pharmaceuticals-15-00974-f001:**
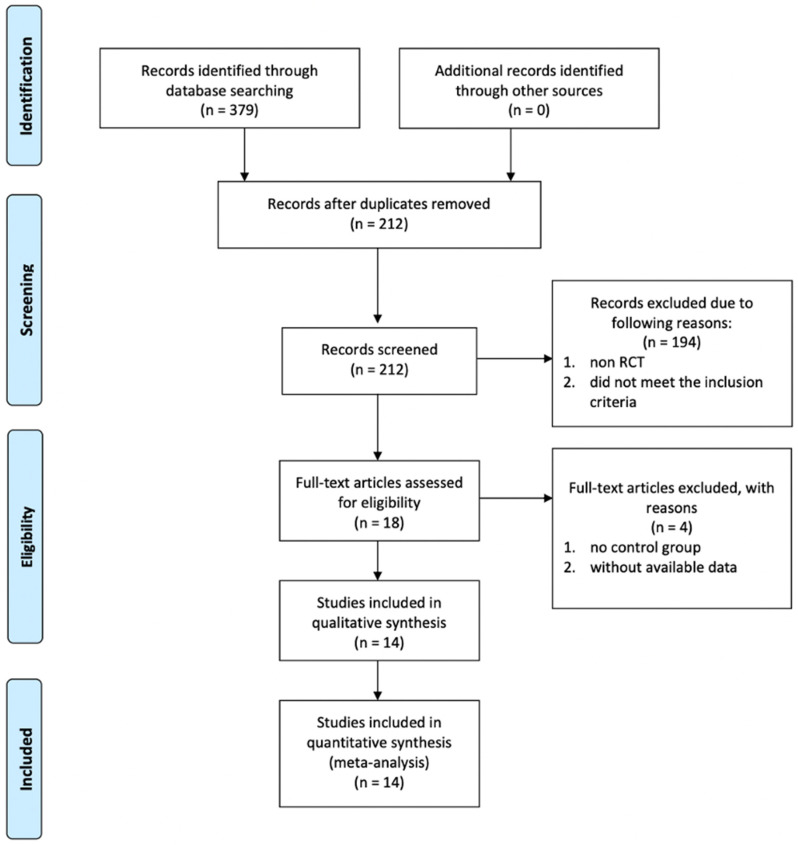
PRISMA flow chart showing literature search and selection process.

**Figure 2 pharmaceuticals-15-00974-f002:**
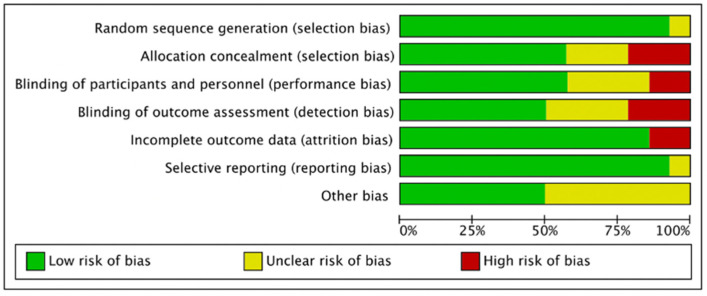
Graph on the risk of bias.

**Figure 3 pharmaceuticals-15-00974-f003:**
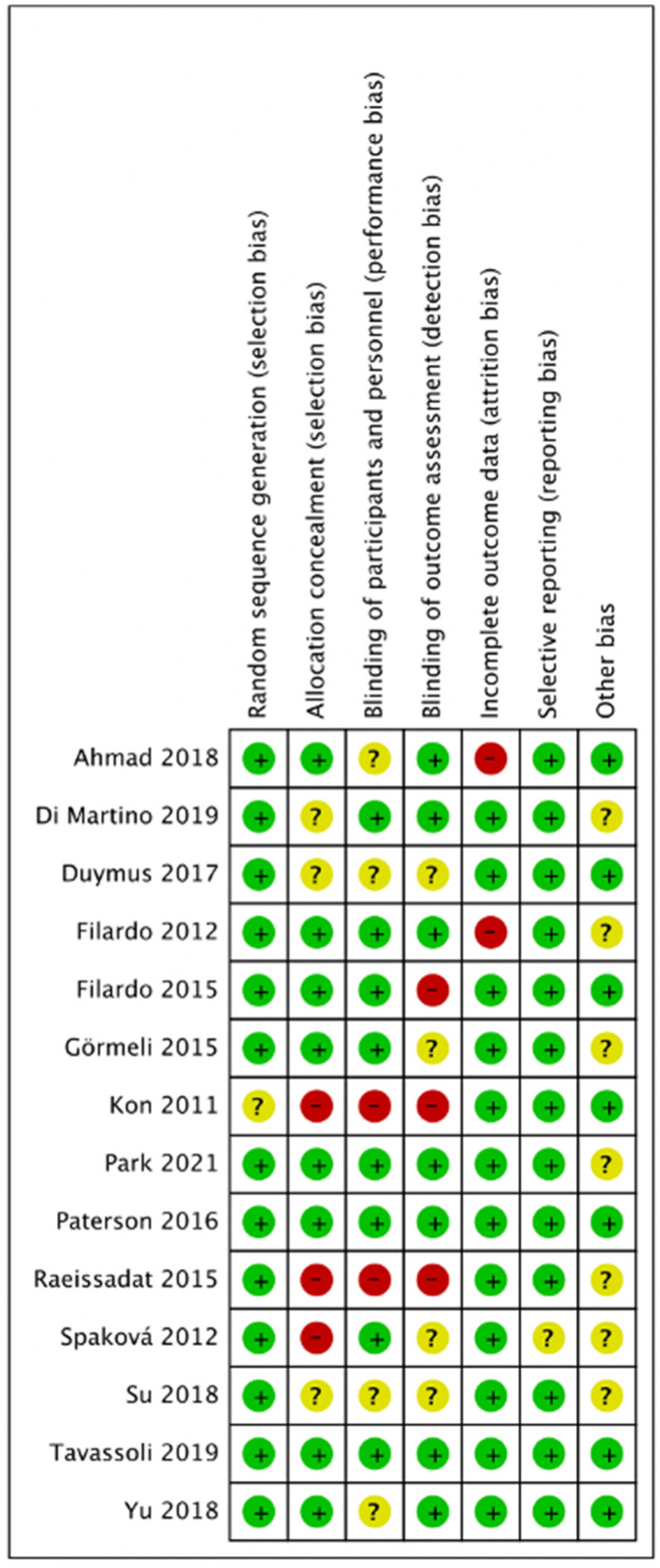
Summary of risk bias.

**Figure 4 pharmaceuticals-15-00974-f004:**
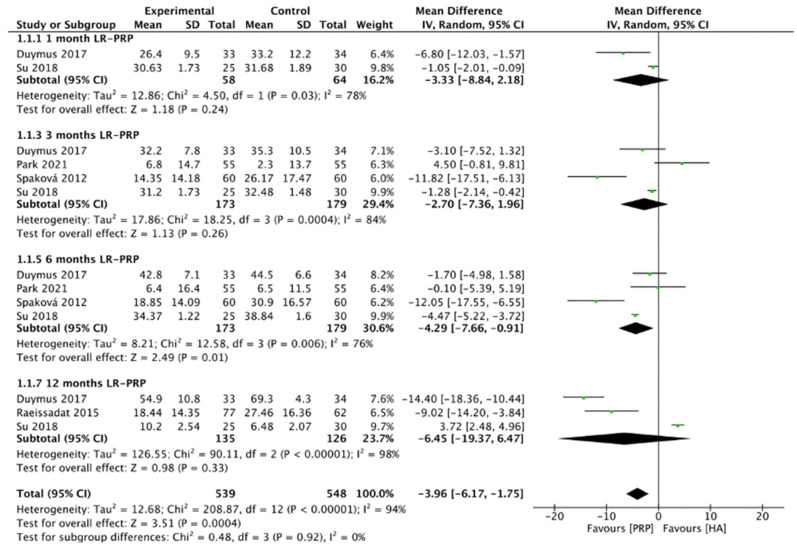
Trials of LR-PRP versus HA. Forest plot of WOMAC total scores. (IV, inverse variance; CI, confidence interval; SD, standard deviation).

**Figure 5 pharmaceuticals-15-00974-f005:**
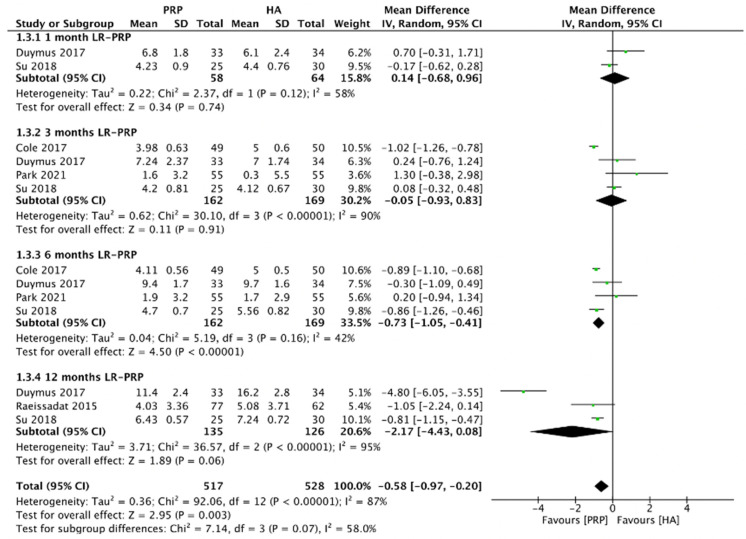
Trials of LR-PRP versus HA. Forest plot of WOMAC pain scores. (IV, inverse variance; CI, confidence interval; SD, standard deviation).

**Figure 6 pharmaceuticals-15-00974-f006:**
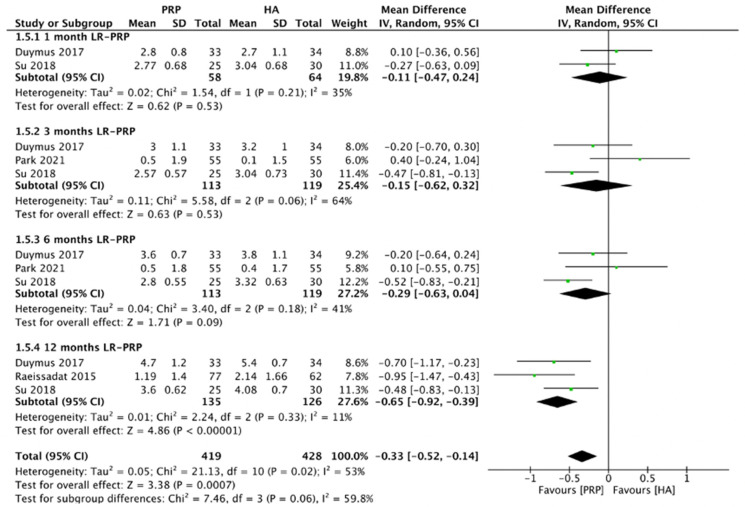
Trials of LR-PRP versus HA. Forest plot of WOMAC stiffness scores. (IV, inverse variance; CI, confidence interval; SD, standard deviation).

**Figure 7 pharmaceuticals-15-00974-f007:**
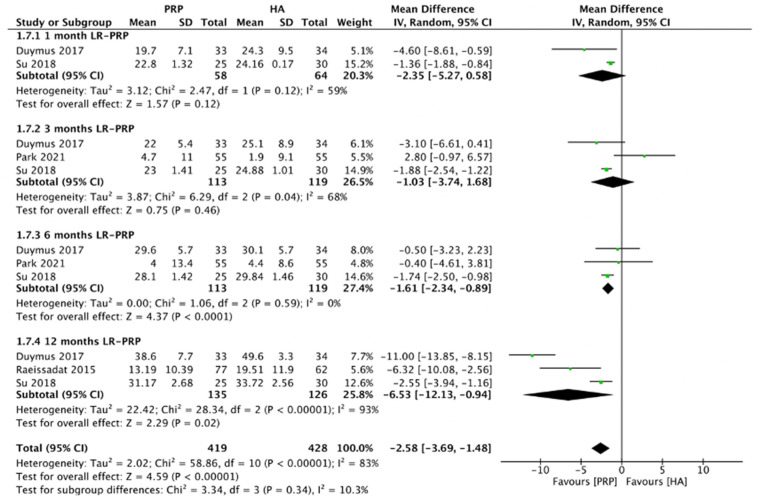
Trials of LR-PRP versus HA. Forest plot of WOMAC physical functional scores. (IV, inverse variance; CI, confidence interval; SD, standard deviation).

**Figure 8 pharmaceuticals-15-00974-f008:**
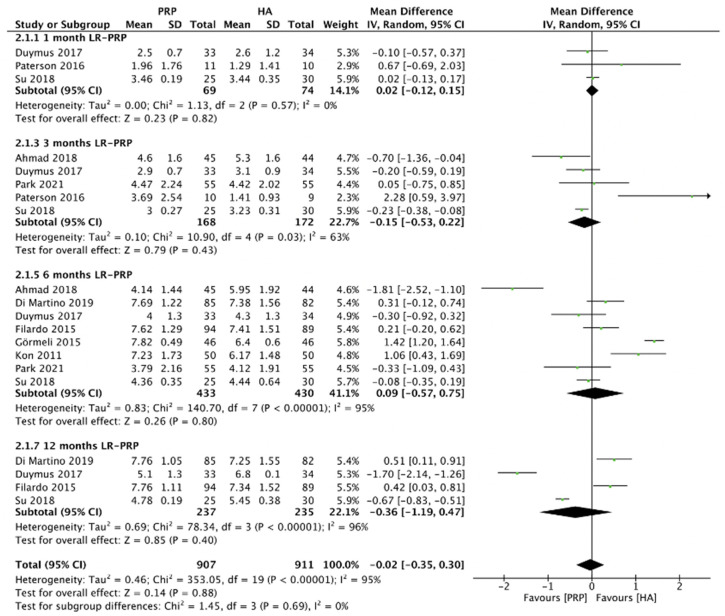
Trials of LR-PRP versus HA. Forest plot of VAS scores. (IV, inverse variance; CI, confidence interval; SD, standard deviation).

**Figure 9 pharmaceuticals-15-00974-f009:**
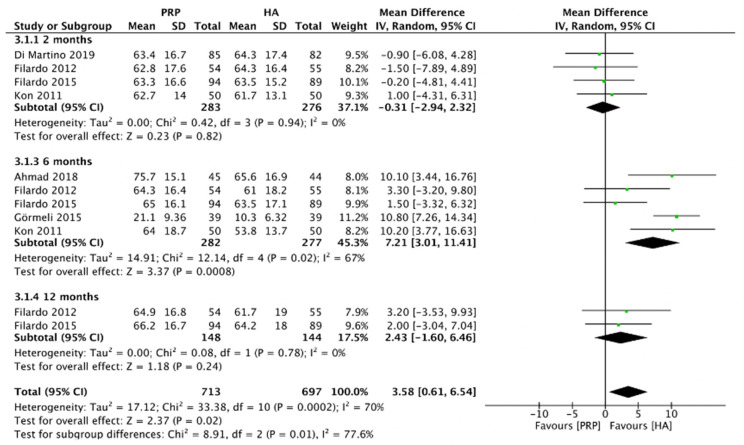
Trials of LR-PRP versus HA. Forest plot of IKDC scores. (IV, inverse variance; CI, confidence interval; SD, standard deviation).

**Figure 10 pharmaceuticals-15-00974-f010:**
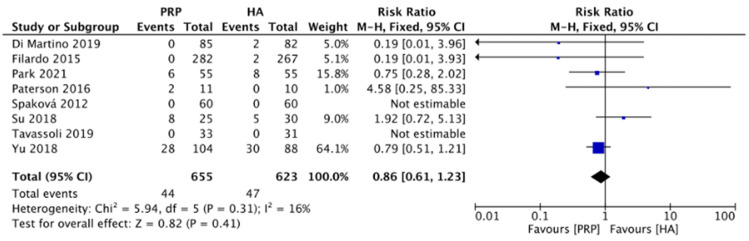
Trials of LR-PRP versus HA. Forest plot of adverse effects. (M-H: Mantel Haenszel; CI, confidence interval).

**Table 1 pharmaceuticals-15-00974-t001:** General characteristics of the included studies.

Included Trials	Study Type	Group	Patients	Age (Years, PRP/HA)	Gender (Male/Female, N)	Stage of Knee OA (Kellgren and Lawrence Classification)	Outcome Measurements	Follow-Up Period (Months)
Kon et al., 2011 (Italy) [11]	RCT	LR-PRP HA	50 50	50.6 ± 13.8 54.9 ± 12.6	30/20 25/25	0–4	EQ-VAS, IKDC, adverse events	2, 6
Filardo et al., 2012 (Italy) [9]	RCT	LR-PRP HA	54 55	55 58	37/17 31/24	0–3	IKDC	2, 6, 12
Spaková et al., 2012 (Slovakia) [12]	RCT	LR-PRP HA	60 60	52.80 ± 12.43 53.20 ± 14.53	33/27 31/29	1–3	WOMAC, adverse events	3, 6
Filardo et al., 2015 (Italy) [13]	RCT	LR-PRP HA	94 89	53.32 ± 13.2 57.55 ± 11.8	60/34 52/37	0–3	EQ-VAS, IKDC, adverse events	2, 6, 12
Görmeli et al., 2015 (Turkey) [14]	RCT	LR-PRP HA	39 39	53.8 ± 23.1 53.5 ± 14	16/23 17/22	0–4	EQ-VAS, IKDC	6
Raeissadat et al., 2015 (Iran) [15]	RCT	LR-PRP HA	77 62	56.85 ± 9.13 61.13 ± 7.48	8/69 15/47	1–4	WOMAC	12
Paterson et al., 2016 (Australia) [16]	RCT	LR-PRP HA	11 10	49.91 ± 13.72 52.70 ± 10.30	8/3 7/3	2–3	VAS, adverse effects	1, 3
Duymus et al., 2017 (Turkey) [17]	RCT	LR-PRP HA	33 34	60.4 ± 5.1 60.3 ± 9.1	1/32 1/33	2–3	WOMAC, VAS	1, 3, 6, 12
Ahmad et al., 2018 (Egypt) [18]	RCT	LR-PRP HA	45 44	56.2 ± 6.8 56.8 ± 7.4	14/31 14/30	1–3	VAS, IKDC	3, 6
Su et al., 2018 (China) [19]	RCT	LR-PRP HA	25 30	54.16 ± 6.56 53.13 ± 6.41	11/14 12/18	2–3	WOMAC, VAS, adverse effects	1, 3, 6, 12, 18
Yu et al., 2018 (China) [20]	RCT	LR-PRP HA	104 88	46.2 ± 8.6 51.5 ± 9.3	50/54 48/40	Not specified	WOMAC, adverse effects	12
Di Martino et al., 2019 (Italy) [10]	RCT	LR-PRP HA	85 82	52.7 ± 13.2 57.5 ± 11.7	53/32 47/35	0–3	EQ-VAS, IKDC, adverse events	2, 6, 12, 24
Tavassoli et al., 2019 (Italy) [21]	RCT	LR-PRP HA	28 27	66.04 ± 7.58 63.30 ± 8.87	6/22 8/19	Ahlbäck classification:1–5	WOMAC, VAS, adverse effects	1, 2, 3
Park et al., 2021 (South Korea) [22]	RCT	LR-PRP HA	55 55	60.6 ± 8.2 62.3 ± 9.6	16/39 8/47	1–3	WOMAC, VAS, adverse effects	1.5, 3, 6

**Table 2 pharmaceuticals-15-00974-t002:** The treatment protocols of IA LR-PRP and IA HA injections.

Included Trials	LR-PRP				HA		
	Dosage (mL)	Intervals (Weeks)	Injection Times	Dosage	Molecular Weight (kDa)	Intervals (Weeks)	Injection Times
Kon et al., 2011 (Italy) [11]	5	2	3	30 mg/2 mL	500–2900	NA	1
Filardo et al., 2012 (Italy) [9]	5	1	3	30 mg/2 mL	>1500	1	3
Spaková et al., 2012 (Slovakia) [12]	3	1	3	NA	Not mentioned	1	3
Filardo et al., 2015 (Italy) [13]	5	1	3	30 mg/2 mL	>1500	1	3
Görmeli et al., 2015 (Turkey) [14]	5	1	3	30 mg/2 mL	Not mentioned	1	3
Raeissadat et al., 2015 (Iran) [15]	4–6	4	2	20 mg/2 mL	500–730	1	3
Paterson et al., 2016 (Australia) [16]	3	1	3	3	Not mentioned	1	3
Duymus et al., 2017 (Turkey) [17]	5	4	2	40 mg/2 mL	1600	4	1
Ahmad et al., 2018 (Egypt) [18]	4	2	3	20 mg/2 mL	Not mentioned	2	3
Su et al., 2018 (China) [19]	6	2	2	2 mL	600–1500	1	5
Yu et al., 2018 (China) [20]	2–14	1	4	0.1–0.3 mg	Not mentioned	1	4
Di Martino et al., 2019 (Italy) [10]	5	1	3	30 mg/2 mL	>1500	1	3
Tavassoli et al., 2019 (Italy) [21]	4–6	3	2	30 mg/2 mL	500–730	1	4
Park et al., 2021 (South Korea) [22]	3	NA	1	30 mg/3 mL	>10,000	NA	1

## Data Availability

Data sharing not applicable.

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
