# Peer review of "Intra-Articular Leukocyte-Rich Platelet-Rich Plasma versus Intra-Articular Hyaluronic Acid in the Treatment of Knee Osteoarthritis: A Meta-Analysis of 14 Randomized Controlled Trials"

_pharmaceuticals, 2022, doi:10.3390/ph15080974_

Round 1
Reviewer 1 Report
Dear authors, I want to congratulate you for this intensive review and meta-analysis. The study is well designed and provides good clinical information to the clinician.
I have some recommendations:
1) Please describe the molecular weight of the hyaluronic acid used in the RCTs?
2) Please comment on the difference between three vs. two injections of LR-PRP. Did you find any difference?
3) Please describe the stages of of knee OA included in the studies
Author Response
Dear Editor and Reviewers August 4th, 2022
This is Dr Carl Chen, the corresponding author to the meta-analysis paper of:
Leukocyte-Rich Platelet-Rich Plasma Versus Intra-Articular Hyaluronic Acid for the Treatment of Knee Osteoarthritis: A Systematic Review and Meta-analysis of 14 Randomized Controlled Trials
We really want to thank the journal and the reviewers for taking the time in revising our manuscript. Both reviewers provided precious suggestions that can make the manuscript more presentable. Therefore, we have done our best to revise the manuscript as suggested by the reviewers. We sincerely hope that the journal can find our revisions satisfactory. It will be our utmost honor to have this manuscript accepted and published by the renowned Pharmaceuticals.
This manuscript was also corrected and read by someone acquainted with English for spelling and grammar mistakes. We have rephrased the sentences, and even amend the title of this manuscript to avoid plagiarism. We will do our best to revise the manuscript again in the future if needed. We believe that the findings discovered in our work can contribute positively to the field of knee osteoarthritis treatment. After reading our work, physicians can have a clearer understanding about plasma rich plasma treatments. In the revised paper draft, all the newly added texts are in red color. All the deleted texts are marked in red color strikeouts. Please feel free to contact me at anytime if I can be of any further assistance. Thank you very much!
Sincerely yours,
Carl P.C. Chen, Professor
Department of Physical Medicine & Rehabilitation
Chang Gung Memorial Hospital, Taiwan
Responses to Reviewer 1:
Dear authors, I want to congratulate you for this intensive review and meta-analysis. The study is well designed and provides good clinical information to the clinician.
- Thank you reviewer 1 for the wonderful comments.
I have some recommendations:
- Please describe the molecular weight of the hyaluronic acid used in the RCTs?
- The molecular weights of the hyaluronic acid that are used in the treatment of knee OA are low and high molecular weights. These molecular weights are now mentioned in Table 2.
- Please comment on the difference between three vs. two injections of LR-PRP. Did you find any difference?
- In most of the references that we have searched, most studies conducted 3 consecutive PRP injections. Some studies received 1, 2, and even 4 injections of PRP. Results in these studies have indicated some degree of effectiveness. However, it is difficult to know the significant differences between these studies.
- We have now mentioned this in the manuscript:
When it comes to the frequency of PRP treatment, most of the included trials received 3 consecutive LR-PRP injections [8-13,15,17]. In further literature search, we have discovered 4 studies that received 2 LR-PRP injection treatments [14,16,18,20]. There was one study that received 4 consecutive LR-PRP injections [19]. One study has mentioned the treatment effectiveness after receiving only 1 LR-PRP injection [21]. Due to the insufficient number of trials, it is difficult to perform subgroup analysis to further arrive at the conclusion as to which frequency can offer the best treatment outcome.
3) Please describe the stages of knee OA included in the studies
We want to thank reviewer 1 for this precious comment. We now added the stages of knee OA in table 1.
Reviewer 2 Report
Overall recommendation: Accept with minor changes
Please add the following:
1. It might be worth adding a paragraph on the types of PRP apart from the leukocyte rich PRP in the introduction.
2. Please re write section 5 as Limitations and conclusions and add the paragraph on the limitations (line 272-280) of your study here.
3. Please expand on the conclusions – reiterating the purpose of your analysis in the article and what you think your article provides that is novel and unique.
4. Please proof read the paper for minor grammatical/English improvements
Author Response
Dear Editor and Reviewers August 4th, 2022
This is Dr Carl Chen, the corresponding author to the meta-analysis paper of:
Leukocyte-Rich Platelet-Rich Plasma Versus Intra-Articular Hyaluronic Acid for the Treatment of Knee Osteoarthritis: A Systematic Review and Meta-analysis of 14 Randomized Controlled Trials
We really want to thank the journal and the reviewers for taking the time in revising our manuscript. Both reviewers provided precious suggestions that can make the manuscript more presentable. Therefore, we have done our best to revise the manuscript as suggested by the reviewers. We sincerely hope that the journal can find our revisions satisfactory. It will be our utmost honor to have this manuscript accepted and published by the renowned Pharmaceuticals.
This manuscript was also corrected and read by someone acquainted with English for spelling and grammar mistakes. We have rephrased the sentences, and even amend the title of this manuscript to avoid plagiarism. We will do our best to revise the manuscript again in the future if needed. We believe that the findings discovered in our work can contribute positively to the field of knee osteoarthritis treatment. After reading our work, physicians can have a clearer understanding about plasma rich plasma treatments. In the revised paper draft, all the newly added texts are in red color. All the deleted texts are marked in red color strikeouts. Please feel free to contact me at anytime if I can be of any further assistance. Thank you very much!
Sincerely yours,
Carl P.C. Chen, Professor
Department of Physical Medicine & Rehabilitation
Chang Gung Memorial Hospital, Taiwan
Responses to Reviewer 2:
- It might be worth adding a paragraph on the types of PRP apart from the leukocyte rich PRP in the introduction.
- Thank you for this import comment. A paragraph is now added in the Discussion section to discuss the types of PRP:
Autologous platelet-rich plasma (PRP) can be divided into leukocyte-rich PRP (LR- PRP) and leukocyte-poor PRP (LP-PRP). By definition, LR-PRP is regarded as having a neutrophil concentration above baseline. LP-PRP is regarded as having a neutrophil concentration below baseline[26] . Present understanding is that LR-PRP may be associated with pro-inflammatory effects. Catabolic cytokines such as interleukin-1β (IL-1β), tumor necrosis factor-α, and metalloproteinases are increased in LR-PRP, which may have deleterious effects on chondrocytes.
- Please re write section 5 as Limitations and conclusions and add the paragraph on the limitations (line 272-280) of your study here.
- Thank you. Section 5 is now Limitations and Conclusions.
- Please expand on the conclusions – reiterating the purpose of your analysis in the article and what you think your article provides that is novel and unique.
- The conclusion is now expanded:
In conclusion, this is the first meta-analysis which demonstrated that IA LR-PRP injection showed better overall outcomes as compared to IA HA injection in patients with knee OA at 3, 6, and 12-month follow-up periods in terms of WOMAC pain scores, WOMAC physical function scores, WOMAC total scores, and IKDC scores. LR-PRP injection offers a better overall treatment outcome as compared with HA in knee OA patients. As a result, IA LR-PRP injections offer better outcomes in treating knee OA patients as compared with HA during both short-term and long-term follow-up periods.
- Please proof read the paper for minor grammatical/English improvements
- Thank you for the significant comment from reviewer 2. We have asked some acquainted with the English language to read through this paper and check for spelling and grammar etc… We have also done our best to delete and to rewrite the content in order to avoid plagiarism. The title of the manuscript is now also changed to avoid possible similarity as compared with other publications.